# Childhood Acute Illness and Nutrition (CHAIN) Network: a protocol for a multi-site prospective cohort study to identify modifiable risk factors for mortality among acutely ill children in Africa and Asia

The Childhood Acute Illness and Nutrition Network

**Correspondence to**
Judd Walson; walson@uw.edu

## ABSTRACT

**Introduction** Children admitted to hospitals in resource-poor settings remain at risk of both inpatient and post-discharge mortality. While known risk factors such as young age and nutritional status can identify children at risk, they do not provide clear mechanistic targets for intervention. The Childhood Acute Illness and Nutrition (CHAIN) cohort study aims to characterise the biomedical and social risk factors for mortality in acutely ill children in hospitals and after discharge to identify targeted interventions to reduce mortality.

**Methods and analysis** The CHAIN network is currently undertaking a multi-site, prospective, observational cohort study, enrolling children aged 1 week to 2 years at admission to hospitals at nine sites located in four African and two South Asian countries. The CHAIN Network supports the sites to provide care according to national and international guidelines. Enrolment is stratified by anthropometric status and children are followed throughout hospitalisation and for 6 months after discharge. Detailed clinical, demographic, anthropometric, laboratory and social exposures are assessed. Scheduled visits are conducted at 45, 90 and 180 days after discharge. Blood, stool and rectal swabs are collected at enrolment, hospital discharge and follow-up. The primary outcome is inpatient or post-discharge death. Secondary outcomes include readmission to hospital and nutritional status after discharge. Cohort analysis will identify modifiable risks, children with distinct phenotypes, relationships between factors and mechanisms underlying poor outcomes that may be targets for intervention. A nested case–control study examining infectious, immunological, metabolic, nutritional and other biological factors will be undertaken.

**Ethics and dissemination** This study protocol was reviewed and approved primarily by the Oxford Tropical Research Ethics Committee, and the institutional review boards of all partner sites. The study is being externally monitored. Results will be published in open access peer-reviewed scientific journals and presented to academic and policy stakeholders.

**Trial registration number** NCT03208725.

## Strengths and limitations of this study

► The Childhood Acute Illness and Illness (CHAIN) cohort is a multi-country study that collects comprehensive data on clinical, laboratory, social, economic and behavioural exposures at multiple sites in Africa and Asia.

► Heterogeneity across sites (geography, rural/urban, varying HIV and malaria prevalence) increases generalisability and may help identify context-independent and dependent interventions.

► Strict harmonisation of procedures, processes and tools allows for comparability between sites.

► This study is deliberately focused on hospitalised children and is not designed to evaluate factors associated with prehospital care and timing of presentation to hospital.

► Bias may be introduced by loss-to-follow-up after hospital discharge if this occurs in more vulnerable children, or if participation in the study alters outcomes.

## INTRODUCTION

Despite impressive global reductions in child mortality, more than five million children under age of 5 die each year.[1] These deaths occur disproportionately in low-income and middle-income countries (LMICs).[2] Factors underlying mortality risk are complex and overlapping, including undernutrition, acute and chronic co-morbidities, lack of access to care, poor social support systems and poverty.[3 4] For many vulnerable children, these factors contribute to an unstable health trajectory, with repeated episodes of illness interspersed with short periods of incomplete recovery (figure 1). In most areas of the world, acutely ill children may not access healthcare during episodes of illness.[5]

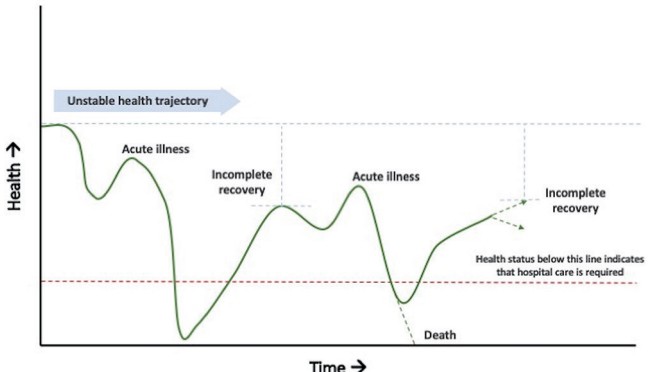

**Figure 1** A model of unstable health trajectory of a child characterised by repeated illness and incomplete recovery.

Of those that do, studies have shown that children who are hospitalised have a dramatically higher risk of death both during hospitalisation and in the months following discharge than their community peers, even when guidelines for treatment and clinical follow-up are followed.[6–9] Thus, contact with the health system resulting in hospitalisation serves as an indicator of vulnerability and a time point where children are readily accessible for intervention.[10–13] While many interventions may be available in hospitals, there are no clear interventions proven to reduce post-discharge mortality. Undernutrition, HIV infection at a young age and a history of repeated hospitalisation have emerged as important risk factors for post-discharge death.[3 9 14]

Approximately half of all childhood mortality is associated with one or more forms of undernutrition, typically identified using simple anthropometry.[2] Mortality rates among children with severe acute malnutrition (SAM) are markedly elevated compared with the general population,[15] but with effective treatment in the community without infectious complications, case fatality may be as low as 1%–2%.[16 17] However, when children with SAM are acutely ill, inpatient case fatality is often 15%–20% in resource-poor countries, with many additional deaths occurring after discharge, even among children who appear to be recovering anthropometrically.[18] Although case fatality rates for children with moderate acute malnutrition (MAM) or stunting in hospital and after discharge have not been widely reported, MAM and stunting are more prevalent than SAM and may contribute to a greater number of child deaths.[2 19]

Current guidelines for the clinical management of acutely ill children with SAM, MAM or stunting are not supported by strong evidence.[20–23] In addition, while measures of mid-upper arm circumference, weight-for-length and length-for-age can identify children as malnourished, anthropometry does not identify the mechanisms that link undernutrition to mortality nor its underlying causes. In typical settings outside humanitarian emergencies, it is unclear to what extent malnutrition is driven by factors other than food intake, including recurrent or chronic infection, intestinal or

systemic inflammation, malabsorption, disability, social vulnerability, parental ill-health, maternal education and agency, lack of financial resources, poor social support and limited access to healthcare.[24] Identifying markers of vulnerability and the underlying mechanisms that drive mortality may offer opportunity to target interventions to improve outcomes in many low-resource settings. In addition, understanding relationships between biological and social risk factors may allow optimisation of intervention packages within specific groups to achieve the largest mortality reductions.

The Childhood Acute Illness and Nutrition (CHAIN) Network is a collaborative group of investigators working across nine sites in four African and two South Asian countries committed to improving child survival and optimising growth and development outcomes in low-resource settings. CHAIN is conducting a prospective cohort study with the objective of determining the phenotypes of groups of children at highest risk, defining the mechanisms underlying such risk and identifying modifiable targets for intervention.

## METHODS
### Objectives
The CHAIN Network is conducting a prospective cohort study of approximately 4500 children aged 1 week to 2 years presenting to healthcare facilities with acute illness deemed severe enough to require admission to hospital at nine sites in six LMICs. Children are followed throughout hospitalisation, discharge and for 6 months post-discharge. The general objective of the study is to characterise acutely ill children and their outcomes in hospital and after discharge in order to identify modifiable pathways leading to death. Using these data, the CHAIN network will prioritise interventions based on deliverability and potential impact.

### Specific objectives
1. To determine factors associated with (1) mortality in hospital; (2) mortality after discharge; (3) readmission to hospital and (4) poor nutritional recovery.
2. To determine differences in clinical, social, and pathophysiological (eg, metabolic, infectious and immune) phenotypes between children of different nutritional status at admission, discharge and follow-up.
3. To identify modifiable risk factors and potential interventions to reduce mortality among vulnerable children.

The goal of the CHAIN Network is to identify interventions that can reduce child mortality beyond the current recommendations for standard of care. Hence, prior to participant enrolment, all study sites underwent a formal assessment of their capacity and adherence to national and international guidelines, including a survey of human resources, materials, equipments, pharmaceutical stock and laboratory capacity. All sites were provided with support to ensure that the minimum standard of care

achieved was in line with the national and international care guidelines. All sites participated in an audit and feed-back cycle of guideline adherence using the individual patient data collected by the CHAIN.

## Locations

Participating sites include rural and urban hospitals in Bangladesh (icddr,b Dhaka Hospital and Matlab Hospital), Burkina Faso (Banfora Regional Referral Hospital), Kenya (Kilifi County Hospital, Mbagathi Sub-County Hospital and Migori County Referral Hospital), Malawi (Queen Elizabeth Central Hospital), Pakistan (Civil Hospital, Karachi), and Uganda (Mulago Hospital).

## Patient and public involvement

The investigators who form the CHAIN Network leadership have extensive experience in the design and conduct of trials focused on improving growth and survival in children in Africa and Asia. The study was designed specifically to reflect patient and public priorities (child survival, reductions in hospitalisation and improved growth). Working closely with Community Advisory Boards at each site, as well as with a robust social science team of researchers in Kenya and Bangladesh, patient priorities, experience and preferences were carefully considered in the design and development of the protocol and data collection instruments. The research questions were also discussed with a number of internal CHAIN Network and external experts in child survival at a convention organised by the funder. Caregivers were included in the Community Advisory Boards to provide input and into the conduct of the study. At the conclusion of the study, results will be disseminated to participants via seminars and outreach events led by the site Principle Investigators and the Community Advisory Boards at each site.

## Eligibility

Children aged 7 days to 23 months who are admitted to hospital with an acute illness are eligible for enrolment. Prior to October 2018, the lower age limit of enrolment was 60 days and this was subsequently modified to allow the inclusion of younger children.

Exclusion criteria include the following:
► Unable to tolerate oral feeds in the 48 hours prior to the onset of the current acute illness.

► Underlying terminal illness that in the opinion of the treating physician is likely to lead to death within 6 months.
► Diagnosed with a condition that in the opinion of the treating physician is likely to require surgery within 6 months.
► Diagnosed with a chromosomal abnormality (syndromically or genetically diagnosed abnormality).
► Primary reason for admission is trauma or surgery.
► Requiring immediate resuscitation at admission defined by ongoing cardiac or pulmonary arrest or judged to be peri-arrest by the attending physician.
► Previous enrolment in the study.
► Sibling enrolled in the study.
► Caregiver plans to move outside of the hospital catchment area within 6 months.
► Caregiver is unwilling to attend study visits.
► Lack of informed consent.

## Screening

Each site has nominated a day on which weekly recruitment begins. The first eligible child in each enrolment group admitted to the hospital is identified and approached for consent. This is repeated for every eligible child admitted subsequently until weekly targets are met for each enrolment stratum (see table 1). Although this is a pseudo-random approach, all the procedures aim to ensure that the study population adequately represents the spectrum of nutritional states observed over the recruitment period.

Potential participants are identified by clinical staff and brought to the attention of the study team who verify the eligibility. If eligible, the study is explained to caregivers in their primary language and written informed consent to participate is sought. Caregivers who are unable to write are asked to provide a witnessed thumbprint. If consent is obtained, the child is enrolled and a unique study number is allocated. If a child is deemed eligible but is too sick for consent to be immediately sought, study staff obtain verbal assent to collect both research and clinical samples at that time to avoid multiple needle insertions. If a caregiver who gave assent then chooses not to provide full written consent, all research data and samples are destroyed. Children classified as orphans or those living in alternative care homes are eligible for enrolment if an appropriate caregiver is present to provide consent on the child's behalf.

| Table 1 | Enrolment groups by age and nutritional status | | | |
|---|---|---|---|---|
| Group | Target Enrolment | Age >6 months | Age 1–6 months | Age <1 month |
| A | 1800 | MUAC <11.5 cm or kwashiorkor | MUAC <11 cm or kwashiorkor | MUAC <9.5 cm or kwashiorkor |
| B | 1800 | MUAC 11.5 to <12.5 cm | MUAC 11 to <12 cm | MUAC 9.5–10.4 cm |
| C | 900 | MUAC ≥12.5 cm | MUAC ≥12 cm | MUAC ≥10.5 cm |

MUAC, mid-upper arm circumference.

## Enrolment

Enrolment is stratified by nutritional status, aiming at a ratio of 2:2:1 (A: B: C) with group targets of 200 group A: 200 group B: 100 group C per site (table 1).

Mid-upper arm circumference (MUAC) was chosen as the optimal measure for participant selection as it is strongly associated with mortality, captures stunted children, varies less with dehydration than weight-based indices and is easily measured in sick children.[25 26]

## Procedures

All sites completed standardised training on variable definitions, identification of clinical signs, measurement of anthropometry, case report form (CRF) completion and data entry prior to starting the study, and training was repeated regularly. An independent study monitor (WESTAT) was hired to conduct site assessments to ensure harmonisation of study procedures across and between sites, as well as to ensure compliance with regulatory standards. Results from WESTAT's monitoring visits are for internal purposes and will be made available to the principal investigators at each site, the study funder, and the CHAIN Network leadership and coordination teams.

On satisfaction of the inclusion criteria and the completion of informed consent forms, a unique study identifier (ID) is allocated. Baseline data, including demographic and social information, a detailed clinical examination and measurement of vital signs, including pulse oximetry, are collected using the standardised CRF. Anthropometry is performed (head circumference, MUAC, weight and length). At admission, biological samples, including blood (up to 5 mL for research purposes), rectal swabs and faecal samples are obtained. All children are offered provider-initiated counselling and testing for HIV, and a malaria rapid diagnostic test is done. Results of investigations performed for clinical care (complete blood count [CBC], biochemistry, glucose or any other laboratory investigations collected) are abstracted and recorded for study purposes. After treatment is initiated, data on the child's diet, social circumstances and, if the mother is present, maternal mental health screening is undertaken. Other data on maternal characteristics collected include maternal MUAC, height, weight and demographic data. Additional maternal variables are listed in the study's enrolment CRF, which is included as an online supplementary file.

During admission, hospitalised children are reviewed daily and specific clinical features indicating illness progression and treatments are recorded on a structured daily CRF that is entered into the CHAIN database. In the event of death in hospital, a standard mortality audit questionnaire is completed by a designated member of the study team.

At discharge, the same clinical assessment as at admission, including anthropometry, is conducted and blood and faecal sampling are repeated.

## Follow-up procedures

A home visit is conducted within 3 days of discharge. The location of the household is recorded by Global Positioning System (GPS) and CRFs are used to capture information on the number of people living in the homestead, access to clean water and improved sanitation, occupation, household assets, income and food security. Parents and legal guardians are also interviewed about their home and social situation, including challenges experienced when keeping their child and family healthy.

Children are followed-up again at 45, 90 and 180 days after discharge at the study facilities, irrespective of scheduled or unscheduled outpatient visits for medical or nutritional care. A standardised questionnaire ascertaining vital status, care-seeking and rehospitalisation history and recent dietary intake (up to 7 days before contact with the hospital) is collected and anthropometry, rectal swabs, faecal specimens and blood samples are repeated at each follow-up visit. Maternal mental health is screened again at day 45.

Children judged to have significant illness at any follow-up visit or at the home visit are referred to an appropriate hospital, clinic or nutrition programme. The study staff share any test result relevant to the patient's care with the clinical management team and families.

Parents and legal guardians are asked to bring their child to the study clinic if they are concerned that the child is unwell. Financial reimbursement for transport and lost earnings at standard local rates is provided at the clinic visit. Study participants who are readmitted to study hospitals undergo the standard clinical assessment delivered at enrolment. Participants who are admitted to hospitals other than the study site hospital have medical data abstracted onto standardised hospital readmission forms. For deaths occurring outside the hospital, a verbal autopsy to evaluate the cause of death is completed within 28 days of study staff becoming aware of a death, using select questions from the WHO standard verbal autopsy tool.[27]

## Community participants

To establish community norms, 125 children at each site living in the same community as hospitalised participants are recruited as community participants based on the following inclusion criteria: absence of known untreated HIV or tuberculosis (TB); no hospital admission in the 14 days prior to contact with the study team and no previously participation in the study. The exclusion criteria listed for hospitalised children also apply to community participants.

At every site, one in four hospital participants has a child enrolled from their community. The selection of hospitalised participants is either every fourth participant, or, in periods when enrolment in hospital was lower, for example, during healthcare workers strikes, one community participant was retrospectively enrolled for every second hospitalised participant. Community participants are identified randomly from the hospitalised participants'

home; a random number x (1–4) and direction (north, south, east, west) are generated using an online tool prior to visiting the home. Random number selection was done using a web-based application, Random Number Generator/Picker.[28] Once at the home of the hospitalised participant, the research team begin by visiting the xth house in the generated direction and attempt to obtain consent to enrol a child within the eligible age range from that household. If not successful, they continue in the same direction to the next xth house. This is repeated until a child is enrolled. Children with severe illness requiring hospital admission identified during community screening are referred for appropriate care.

When an eligible community participant is identified, their caregiver is given information about the study and invited to the study clinic for assessment. Following confirmation of informed consent, a clinical examination and anthropometry are completed and documented, and blood and stool samples collected as in the hospitalised children. Household and demographic questionnaires are also administered. Children in this group requiring non-urgent medical care receive basic treatment in the study clinic and/or are referred to appropriate treatment centres after enrolment procedures are completed. These children remain eligible for inclusion as community participants. Financial reimbursement for transport and lost earnings at standard local rates is provided to each caregiver at the clinic visit. No follow-up is done on community participants.

### Specimen collection

To ensure comparability, standard operating procedures are followed at each site. Blood samples are collected at enrolment, each day of hospitalisation, in the event of clinical deterioration (defined by the onset of a new

Integrated Management of Childhood Illness danger sign), at discharge, at all follow-up points and at any hospital readmission. Blood is collected into a BD Vacutainer Hemogard $K_2$EDTA (spray dried) and a red top with a clot activator (spray dried) at each time point and three dried blood spots on Whatman filter paper cards are prepared. At each time point, a CBC and clinical biochemistry, including sodium, potassium, calcium, magnesium, urea, creatinine, albumin, bilirubin, alanine aminotransferase, inorganic phosphate and alkaline phosphate are performed. Blood glucose, HIV testing (Alere Determine or Uni-Gold HIV) and malaria rapid diagnostics tests (CareStart HRP2/pLDH) are also performed on all children at enrolment and at additional time points, if clinically indicated. Caregivers may refuse any sample collection during the study without being excluded from further follow-up. The schedule for blood collection is detailed in table 2.

Fresh stool collected directly or from child diapers is transferred into standardised stool collection pots by study staff, aliquoted and stored at −80˚C. Advanced pathogen detection using Taqman Array Card, sequencing, microbiome analysis, metabolomics, and markers of enteric inflammation and dysfunction will be conducted on a subset of stool samples in the CHAIN nested case–control study.

### Data management and confidentiality

Data recorded on standardised CRFs are de-identified and entered in a secured central database and housed on servers in Nairobi with secure offsite backup. Prior to and after data entry, paper records are kept in a locked room with locked filing cabinets at each site, with access limited to investigators and study staff directly involved in data collection and entry. For de-identification, only

**Table 2**  Schedule of blood samples and tests performed at each collection time point

| | | Admission | Discharge | Day 45 | Day 90 | Day 180 | Readmission |
|---|---|---|---|---|---|---|---|
| Results used for care | Complete blood count | X | X | X | X | X | X |
| | Biochemistry | X | | | | | X |
| | Glucose | X | | | | | X |
| | Blood gas+lactate* | X | | | | | X |
| | Blood culture* | X | | | | | X |
| | Malaria Rapid Diagnostic Test (RDT) | X | | | | | X |
| | HIV test | X | | | | | |
| Processed or stored for research assays only | Dried blood spot | X | X | X | X | X | X |
| | Serum+plasma for storage | X | X | X | X | X | X |
| | Whole blood for storage | X | X | X | X | X | X |
| | Functional immunology* | X | X | X | X | X | X |
| | Peripheral Blood Mononuclear Cell (PBMC) extraction* | X | X | X | X | X | X |
| | Rectal swab and whole stool | X | X | X | X | X | X |

*Done at a subset of sites with capacity, thus these are sub-studies.

participant initials are recorded and any potential identifier such as date of birth and GPS location are stored separately. Site principal investigators or their delegates conduct regular internal quality assurance on completion, data transfer and storage of the Case Reporting Forms (CRFs). A central data management system generates automated queries and reviews incoming data weekly. Any missing or implausible data are queried and site teams are given specific timeframe to resolve the raised issues.

## Analysis

Data from all Network sites will be combined as a single cohort. Outcomes will be classified into one or more of the following, the proportion of children with each endpoint described with 95% CIs:

► Death at any time during the period of observation (primary endpoint).
► Death in hospital.
► Death after discharge.
► Readmission to hospital within 180 days.
► Nutritional status at 180 days.
► Recovery as defined by survival without readmission or development of SAM during 180 days after discharge.

### Primary cohort analysis—mortality

The hazard of mortality during the study will be calculated using multivariable survival models including (1) clinical signs and symptoms; (2) anthropometric markers of wasting and stunting; (3) markers of organ function; (4) birth history and prior health and (5) maternal, household and social factors. The stratified design by location and nutritional category will be accounted for with multilevel fixed and random effects. Subgroup analyses will be conducted for inpatient and post-discharge mortality, including time-to-event survival analyses. Predictive algorithms will be built using methods such as classification and regression trees, boosted regression trees and other machine learning methods. Mechanistic pathways will be interrogated using structured equation modelling, latent class analysis and other methods.

### Secondary cohort analysis

Secondary endpoints are rehospitalisation and the presence of SAM or MAM after 6 months. These will be examined using generalised linear models accounting for the competing risk of mortality. Growth trajectories post-discharge will be compared with WHO standard growth curves and modelled in relation to risk factors and outcomes as panel data. Differences in mortality between sites will be examined in relation to case-mix and underlying risk factors such as HIV and socioeconomic factors.

### Nested case–control study

A nested case–control design will be utilised to investigate mechanisms with advanced laboratory testing on stored samples for efficiency. Cases will be defined as (1) children who died during follow-up and (2) children who are readmitted to hospital. Controls will be selected from children who survive to 6 months without readmission ('pure controls'), matched by site and nutritional strata to reflect the study design.

Primary analyses will estimate the association between the exposures of interest and the odds of poor outcomes during follow-up. The exposures of interest in these analyses will include plasma and faecal markers of intestinal dysfunction, systemic inflammation, systemic and enteric pathogens, 'maturity' and diversity of the enteric microbiome, proteomic and metabolomic markers of energy metabolism, and macronutrient and micronutrient status.

If mortality in the study is lower than expected (500 observed deaths), the primary case definition will be expanded to include children who were enrolled but were rehospitalised post-discharge. Samples from these children may also be used in a sensitivity analysis to determine if associations with readmission with severe illness ('near-miss') differ from those for mortality.

## Sample size

Sample size calculations are based on expected events (deaths) in the cohort. We anticipate based on prior information from the sites that recruiting approximately 4500 children in the specified strata will result in 500 deaths in this population. The sample size estimation is based on detecting differences in the proportion who die post-discharge between moderately malnourished and non-malnourished groups. Among non-malnourished children, we assumed an inpatient case fatality proportion of 5.0%, a cumulative post-discharge mortality of 2.5% compared with 7.5% inpatient case fatality among moderately malnourished children and allowing for 10% loss to follow-up. For a two-sided hypothesis that Ha: $p2 \neq p1$, with an $\alpha$ of 0.05, a power of 80% is attained for a post-discharge case fatality of 4.8% or above.

For laboratory analyses on stored samples, a nested case–control approach will compare cases (children who die) and controls (children with good recovery) in a 1:2 ratio matched by site and nutritional strata. We expect approximately 500 deaths will occur, with 1000 controls. For a two-sided hypothesis that Ha: $p2 \neq p1$ at 80% power and a significance level of 0.05, this allows for the determination of an OR of 1.8 for risk factors with a prevalence of 5% among controls, and of 1.4 for risk factors with a prevalence of 25%. Further analyses using a combined secondary endpoint of death and/or rehospitalisation will ensure adequate power to detect risk factors with lower prevalence.

At each site, inclusion of 125 community participants will permit the calculation of descriptive percentages as integer values and estimates of community norms of continuous variables.[29] Across the whole study, 1025 community participants will be recruited. For a two-sided hypothesis that Ha: $p2 \neq p1$, at 80% power and a significance level of 0.05 and assuming clustering by site, this sample size allows for the determination of a prevalence

ratio of 1.5 for risk factors with a prevalence of 5% among community participants and a ratio of 1.25 for risk factors with a prevalence of 25%.

## Study timeline

The CHAIN Cohort Study began enrolling on 20th November 2016 and participant recruitment and follow-up is expected to occur through August 2019.

## Potential challenges and limitations

The study is designed and powered to detect associations with mortality, the primary outcome. Prior data from other studies conducted at the CHAIN sites suggest that sufficient numbers of events will occur to achieve adequate power. However, an interim analysis will be conducted in 2019 to confirm that the cohort study is adequately powered to detect the effects of covariates of interest. In the event that mortality or enrolment rates are not sufficient, we will include rehospitalisations as a combined primary endpoint. This cohort study is being conducted in low-resource environments where the risk of civil, political or military disruption, and industrial action affecting hospitals are significant. The inclusion of multiple sites across a wide geographical range allows for some sites to strategically increase enrolment if other sites are unable to achieve planned targets.

Bias may occur if children lost to follow-up are not representative of the study population; it is anticipated that non-attenders may be more vulnerable. There is also the risk of the Hawthorne effect where involvement in the study alters outcomes.

## Ethics and dissemination

Prior to project inception, the key stakeholders at each site were engaged, including those from relevant Ministries of Health, local academic institutions, hospitals hosting the study and the community engaged in the research. Community Advisory Boards have been assembled at each site. Study progress and results are shared with the key stakeholders as well as disseminated through workshops and written materials. The CHAIN Network has both ethics and policy advisory boards to providence guidance on how to tailor research activities and help disseminate key findings with enough reach and power to influence high-level policy decisions.

**Acknowledgements** The CHAIN Network thanks all the patients and their families for participating in this study. We acknowledge all leadership and staff at CHAIN hospital sites, and especially the clinical staff for sharing their resources and time to both care for CHAIN patients and collect pertinent data for the study.

**Collaborators** Berkley, JA (KEMRI/Wellcome Trust Research Programme, Kilifi, Kenya; Center for Tropical Medicine and Global Health, University of Oxford, Oxford, UK; Network PI). Walson, JL (Departments of Global Health, Medicine, Pediatrics and Epidemiology, University of Washington, Seattle, Washington, USA; Network PI). Diallo, AH (Department of Public Health, Faculty of Health Sciences, University of Ouagadougou). Ki-Zerbo J, Ouagadougou, Burkina Faso; Department of Public Health, Centre Muraz Research Institute, Ministry of Health, Bobo-Dioulasso, Burkina Faso). Shahid, ASMSB (Nutrition and Clinical Services Division (NCSD), International Centre for Diarrhoeal Disease Research, Bangladesh (icddr,b), Dhaka, Bangladesh; Co-investigator). Gwela, A (KEMRI/Wellcome Trust Research Programme, Kilifi, Kenya. Saleem, A (Department of Pediatrics and Child Health, The Aga Khan University,Karachi 74800, Pakistan). Asad, A (Department of Pediatrics and Child Health, The Aga Khan University, Karachi 74800, Pakistan). Tigoi, CC (KEMRI/Wellcome Trust Research Programme, Kilifi,Kenya). Bourdon, C (Department of Translational Medicine, The Hospital for Sick Children, Toronto, Ontario, Canada). Lancioni, CL (Department of Pediatrics, Oregon Health and Science University,Portland, Oregon, USA). Denno, DM (Department of Pediatrics,School of Medicine, University of Washington, Seattle, Washington, USA; Department of Global Health and Department of Health Services, School of Public Health, University of Washington, Seattle, Washington, USA). Mangale, DI (Department of Global Health, University of Washington, Seattle, Washington,USA). Mupere, E (Department of Paediatrics and Child Health,College of Health Sciences, Makerere University, Kampala, Uganda). Tickell, KD (Departments of Global Health and Epidemiology, University of Washington, Seattle, Washington, USA). Mwangome, MK (KEMRI/Wellcome Trust Research Programme, Kilifi, Kenya). Chisti, MJ (Nutrition and Clinical Services Division (NCSD), International Centre for Diarrhoeal Disease Research, Bangladesh (icddr,b), Dhaka,Bangladesh). Ngari, MM (KEMRI/Wellcome Trust Research Programme, Kilifi, Kenya). Ngao, NM (KEMRI/Wellcome Trust Research Programme, Kilifi, Kenya). Sukhtankar, P (KEMRI/Wellcome Trust Research Programme, Kilifi, Kenya). Bandsma, RHJ (Centre for Global Child Health, The Hospital for Sick Children,Toronto, Ontario, Canada; Division of Gastroenterology, Hepatology and Nutrition, The Hospital for Sick Children, Toronto, Ontario, Canada; Department of Nutritional Sciences, Faculty of Medicine, University of Toronto, Toronto,Ontario, Canada; Department of Biomedical Sciences, College of Medicine,University of Malawi, Blantyre, Malawi). Molyneux, S(KEMRI Centre for Geographic Medicine Research — Coast and Wellcome Trust Research Programme, Nairobi, Kenya; Centre for Tropical Medicine, University of Oxford, Oxford, UK). Ahmed, T (Nutrition and Clinical Services Division (NCSD), International Centre for Diarrhoeal Disease Research, Bangladesh (icddr,b), Dhaka, Bangladesh). Voskuijl, W (Department of Paediatrics and Child Health,College of Medicine, University of Malawi, Blantyre, Malawi; Department of Biomedical Sciences, College of Medicine, University of Malawi, Blantyre,Malawi).

**Contributors** JB and JW led the proposal and protocol development with input from all collaborators listed.

**Funding** This work was supported by the Bill and Melinda Gates Foundation, grant number OPP1131320.

**Competing interests** None declared.

**Patient consent for publication** Not required.

**Ethics approval** This study protocol was reviewed and approved by the Oxford Tropical Research Ethics Committee, UK; the Kenya Medical Research Institute, Kenya; the University of Washington and Oregon Health and Science University, USA; Makerere University School of Biomedical Sciences Research Ethics Committee and The Uganda National Council for Science and Technology, Uganda; Aga Khan University, Pakistan; the International Centre for Diarrheal Disease Research, Bangladesh; The University of Malawi; The University of Ouagadougou and Centre Muraz, Burkina Faso; the Hospital for Sick Children, Canada; and University of Amsterdam, The Netherlands.

**Provenance and peer review** Not commissioned; externally peer reviewed.

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
