## [Reviewer comments · BMJ Open]

ARTICLE DETAILS

TITLE (PROVISIONAL)	The Childhood Acute Illness and Nutrition (CHAIN) Network: Protocol for A Multi-site Prospective Cohort Study to Identify Modifiable Risk Factors for Mortality Among Acutely Ill Children in Africa and Asia.
AUTHORS	The Childhood Acute Illness and Nutrition (CHAIN), Network; Walson, Judd

VERSION 1 - REVIEW

REVIEWER	André BRIEND Department of Nutrition, Exercise and Sports, Faculty of Science, University of Copenhagen, Rolighedsvej 30, DK-1958 Frederiksberg, Denmark University of Tampere School of Medicine and Tampere University Hospital, University of Tampere, University of Tampere School of Medicine Center for Child Health Research, Lääkärintäti 1, Arvo building, FI-33014 University of Tampere Tampere, Finland
REVIEW RETURNED	20-Dec-2018

GENERAL COMMENTS	There are many published studies describing factors associated with poor outcomes, mainly with the risk of death among children treated as inpatients for severe acute malnutrition but these studies are usually quite old, they describe children treated with old protocols and it is not sure their findings are still relevant today. These old studies have no community controls, which limits the interpretation of their findings. Also, most of them had a limited follow-up whereas this one will examine what happens to children up to 6 months after being discharged from the hospital. This is highly relevant as these children have an excess mortality several months after being discharged. The proposed study will also look at risk factors which were not looked at in old studies: plasma and faecal markers of intestinal dysfunction, systemic inflammation, systemic and enteric pathogens, 'maturity' and diversity of the enteric microbiome, proteomic and metabolomic markers of energy metabolism, and macro- and micronutrient status. This will help to improve our understanding of the mechanisms leading to a poor outcome in severe acute malnutrition and to improve the treatment of these children. This is a very important and timely study. There is no mention of oedema detection among the admission procedures. Oedema has often been looked at as a possible aggravating factor of severe acute malnutrition but its contribution
---

	is difficult to assess from previous studies as nutritional status was often measured by weight based indices which are influenced by the degree of oedema. This study assessing nutritional status with MUAC will not have this limitation. Little information will be collected about the mother, apart from mental health problems screening. In particular, there is no mention of the mother's nutritional status. Measuring mother's MUAC as a potential risk factor could be considered.
--	---

REVIEWER	Matthew O Wiens University of British Columbia, Canada
REVIEW RETURNED	03-Jan-2019

GENERAL COMMENTS	This manuscript represents the protocol for the CHAIN Cohort Study. This study aims to characterize the biomedical and social risk factors for both in-hospital and post-discharge mortality among acutely ill children, both with and without malnutrition. This study is designed as a three-arm cohort study (eligibility within each arm is based on nutritional status), and is being conducted at 9 hospital sites in six countries. This is an important study that is certain to contribute to a more nuanced understanding of both in-hospital and post-discharge risk. Because of both the importance of the post-discharge period as a critical contributor to childhood morbidity and mortality, and because of the well designed nature of this project, I fully support the publication of this protocol. Although I strongly support this manuscript, there are a few minor points of clarification needed. Furthermore, I am advocating for a more detailed and thoughtful approach in describing the limitations of this study. Screening: The manuscript states that each site nominates a day on which weekly recruitment begins, and that recruitment continues consecutively until weekly targets are met. It is stated that this aims to ensure the study population adequately represents the true spectrum seen over the recruitment period. I'm not sure how this method can do this? This is only possible if targets are generally not met each week. Otherwise, later days of the week are not well represented. Are the targets such that enrollment is representative for each day of the week? What about time of day? Does enrollment occur at night? Does the nominated day of enrollment change over the course of enrollment to ensure balance by day of week? I think that if the authors wish to include the statement about representing the true spectrum of patients further methodological justification is required. Otherwise, perhaps removing this statement about ensuring balance would be appropriate. Procedures: It is stated that an independent study monitor (WESTAT) was hired to conduct site assessments to ensure harmonization across sites and protocol compliance. Perhaps a statement as to how
--

these results will be used and eventually reported would be informative to the reader.

It is stated that clinical staff (rather than study staff) identify potentially eligible children, bringing these children to the attention of the study team. How is this process monitored? Clinical staff have (especially at teaching hospitals) have high rates of turnover. Also, without any incentive, clinical staff may not wish to use their personal resources (ex. airtime) or time to inform study staff. Again, I'm not sure that this method ensures that the study sample is truly representative of the target population. My experience in Uganda suggests that having study staff present on the wards (where recruitment occurs) is the only good method of ensuring that ALL potentially eligible subjects are screened, an important requirement to ensure a representative study sample.

Each discharged child receives a home visit within 3 days of discharge, with what appears to be an extensive interview. While certainly providing vital information, it is possible that this may have the unintended effect of acting like an intervention, thus affecting the outcome. The authors certainly recognize that this visit occurs during the most vulnerable time in the post discharge period. This then becomes less generalizable to the general population, where such a visit is not done.

Providing reimbursement for transport and lost earnings during any follow-up is likely to affect the outcome, and further reduce generalizability. While it is certainly laudable that the research team is able to provide such a high level of support, this is likely to reduce the validity of any factors (modifiable or not) to a more typical/general population where post-discharge mortality contributes substantially to childhood mortality.

Should the statement about the home visit at 3 days post discharge be better placed in the section on follow-up procedures? This does seem to be, ultimately, a component of the follow-up? I do not have a strong opinion about this, it is only a suggestion.

Community Participants:

The first word of the second sentence is not capitalized, and actually appears incomplete.

Potential Challenges and Limitations:

In the limitations section, it states that an interim analysis "will be conducted in 2018". It is now January 2019. Has this analysis been conducted? If so, should any of these results be presented? I am not suggesting that they should necessarily be presented, but perhaps a statement that the interim analysis did suggest that the study was adequately powered would be appropriate.

I think that perhaps more should be said of what the authors suggest could be a Hawthorne effect. I actually believe that this potential bias goes beyond merely an observer effect, where the subject/caregiver behaves differently given that they are being observed. I believe that this study (especially the post-discharge component) indeed acts in many ways like an intervention. Not only is post-discharge health seeking is paid for by the study, but also a payment for lost earnings is provided during all follow-up

	visits. It must be acknowledged that the removal of these common and critical health seeking barriers may affect the very outcome being studied. While internal validity is not likely altered, these results are not likely to truly reflect the context where the primary outcome (mortality - and more specifically post-discharge mortality) is a problem. It is certainly plausible that interventions focusing on an improved discharge process, and improved post-discharge follow-up may indeed be the most important aspects of interventions to improve this critical outcome.
--	--

VERSION 1 – AUTHOR RESPONSE

2. Reviewer 1:

a. There is no mention of oedema detection among the admission procedures. Oedema has often been looked at as a possible aggravating factor of severe acute malnutrition but its contribution is difficult to assess from previous studies as nutritional status was often measured by weight-based indices which are influenced by the degree of oedema. This study assessing nutritional status with MUAC will not have this limitation.

Response: Thank you for your comments. In addition to evaluating anthropometry in all enrolled participants (MUAC, height and weight), all children were also assessed for kwashiorkor by looking for the presence of bipedal oedema. This was part of the criteria used to identify members of group A. The presence of kwashiorkor is highlighted in table 1 on page

6.

b. Little information will be collected about the mother, apart from mental health problems screening. In particular, there is no mention of the mother's nutritional status. Measuring mother's MUAC as a potential risk factor could be considered.

Response: Our apologies for not mentioning this in the paper. Data are collected from all primary caregivers, including maternal MUAC, height, weight and other demographic characteristics. This forms a significant proportion of the social science aspect of the CHAIN study. We have now added mention of maternal variables to the manuscript. Please see page 7 of the protocol. A copy of the enrolment case report form which lists maternal variables collected for this study, is also included in the appendix.

3. Reviewer 2:

a. Screening:

The manuscript states that each site nominates a day on which weekly recruitment begins, and that recruitment continues consecutively until weekly targets are met. It is stated that this aims to ensure the study population adequately represents the true spectrum seen over the recruitment period. I'm not sure how this method can do this? This is only possible if targets are generally not met each week. Otherwise, later days of the week are not well represented. Are the targets such that enrollment is representative for each day of the week? What about time of day? Does enrollment occur at night? Does the nominated day of enrollment change over the course of enrollment to ensure balance by day of week? I think that if the authors wish to include the statement about representing the true

spectrum of patients further methodological justification is required. Otherwise, perhaps removing this statement about ensuring balance would be appropriate.

Response: Thank you for this response. In the development of the protocol, we wanted to ensure that all sites were as closely harmonized in enrolment methodology to ensure both representative enrollment at each site and to ensure that bias in enrolment was not a problem within or between sites. Given that staffing levels vary at each site (as does the ability to enroll at night or on weekends), we chose a pseudo-random selection approach as described. We acknowledge that this is not a perfect method but in reviewing recruitment data to date it appears that enrolment was well balanced across seasons, months and days of the week. Most enrolment was initiated in the morning and early afternoon at most sites given that staff often were not available in the evening or at night to enroll participants. We have added a statement to the protocol noting that this is a pseudo-random selection of participants. This can be found on page 5 of the protocol.

b. Procedures:

i. It is stated that an independent study monitor (WESTAT) was hired to conduct site assessments to ensure harmonization across sites and protocol compliance. Perhaps a statement as to how these results will be used and eventually reported would be informative to the reader.

Response: WESTAT was retained as a study monitor for internal purposes. As this study was not a clinical trial, independent trial monitoring was not necessary. However, as we hope to use the CHAIN Network as a platform for future trials, we worked closely with WESTAT to ensure that the sites were receiving feedback that would enable them to meet future requirements for monitoring. In addition, we specifically requested WESTAT to provide the study leadership with assessments of site harmonization and compliance with study protocols and SOPs for internal monitoring purposes. Results from WESTAT's monitoring were made available to the principal investigators at each site, the study funder and the CHAIN Network leadership and coordination teams. We have added a statement to this effect on page 6.

ii. It is stated that clinical staff (rather than study staff) identify potentially eligible children, bringing these children to the attention of the study team. How is this process monitored? Clinical staff have (especially at teaching hospitals) have high rates of turnover. Also, without any incentive, clinical staff may not wish to use their personal resources (ex. airtime) or time to inform study staff. Again, I'm not sure that this method ensures that the study sample is truly representative of the target population. My experience in Uganda suggests that having study staff present on the wards (where recruitment occurs) is the only good method of ensuring that ALL potentially eligible subjects are screened, an important requirement to ensure a representative study sample.

Response: We thank the reviewer for raising these concerns. The CHAIN teams and clinical staff at each hospital were working side by side at most times, with CHAIN teams often also providing clinical care when sites requested. The CHAIN leadership conducted a clinical site assessment (which will be presented in a future manuscript, in preparation) that evaluated the success of this screening method. All sites developed close relationships between the study staff and the clinical teams which was effective in removing some of the barriers described in the feedback above. Finally, all screening logs were reviewed regularly to ensure that recruitment remained representative of the population of interest.

iii. Each discharged child receives a home visit within 3 days of discharge, with what appears to be an extensive interview. While certainly providing vital information, it is possible that this may have the unintended effect of acting like an intervention, thus affecting the outcome. The authors certainly recognize that this visit occurs during the most vulnerable time in the post discharge period. This then becomes less generalizable to the general population, where such a visit is not done.

Response: We appreciate the comment from the reviewer. We made every effort to minimize unnecessary contact with participants to reduce the potential for Hawthorne effects. However, a home assessment was felt to be necessary for ethical reasons and for data accuracy. We have also found previously that a home visit is beneficial in establishing trust and preventing loss to follow up. We acknowledge that this may affect generalizability of these findings and plan to discuss this in the final manuscript. Such effects are also a reason why readmission to hospital for serious illness is a major outcome in addition to mortality.

iv. Providing reimbursement for transport and lost earnings during any follow-up is likely to affect the outcome, and further reduce generalizability. While it is certainly laudable that the research team is able to provide such a high level of support, this is likely to reduce the validity of any factors (modifiable or not) to a more typical/general population where post-discharge mortality contributes substantially to childhood mortality.

Response: We acknowledge that reimbursements may affect the motivation of caregivers and thereby influence the outcome. Again, we felt that the need to minimize lost-to-follow up outweighed the bias introduced in facilitating access to care at follow up.

Reimbursement was set at the lowest possible rate to meet the transport needs and cover lost earnings for the day of travel to the hospital to minimize effect on the study outcomes. These rates were set after discussion with the Community Advisory Boards at each site.

v. Should the statement about the home visit at 3 days post discharge be better placed in the section on follow-up procedures? This does seem to be, ultimately, a component of the followup? I do not have a strong opinion about this, it is only a suggestion.

Response: Thank you for this suggestion. We have amended the protocol to reflect your recommendation. See page 7.

vi. Community Participants: The first word of the second sentence is not capitalized, and actually appears incomplete.

Response: Thank you for this suggestion. Corrections have been made per your suggestion. See page 7.

c. Potential Challenges and Limitations:

i. In the limitations section, it states that an interim analysis "will be conducted in 2018". It is now January 2019. Has this analysis been conducted? If so, should any of these results be presented? I am not suggesting that they should necessarily be presented, but perhaps a statement that the interim analysis did suggest that the study was adequately powered would be appropriate.

Response: Thank you for noting this error. We have updated the limitations section to reflect the fact that we will be conducting an interim analysis in early 2019. Given the number of events accrued to date, we are confident that the interim analysis will show that the study is adequately powered. See edits on page 12 of the manuscript.

ii. I think that perhaps more should be said of what the authors suggest could be a Hawthorne effect. I actually believe that this potential bias goes beyond merely an observer effect, where the subject/caregiver behaves differently given that they are being observed. I believe that this study (especially the post-discharge component) indeed acts in many ways like an intervention. Not only is post-discharge health seeking is paid for by the study, but also a payment for lost earnings is provided during all follow-up visits. It must be acknowledged that the removal of these common and critical health seeking barriers may affect the very outcome being studied. While internal validity is not likely altered, these results are not likely to truly reflect the context where the primary outcome (mortality -

and more specifically post-discharge mortality) is a problem. It is certainly plausible that interventions focusing on an improved discharge process, and improved post-discharge follow-up may indeed be the most important aspects of interventions to improve this critical outcome.

Response: We acknowledge that CHAIN is a well-resourced trial. The intention is to study mortality despite adherence to current guidelines for care rather than 'usual' care, however we recognize that children in the study may have received a level of access to care that may attenuate risk of mortality. This was a careful decision balancing cross-study and local ethical imperatives with study objectives. We would expect that this may bias our results towards reducing mortality. However, we have seen mortality rates exceeding 10% for the entire cohort, indicating that the study will be more than adequately powered to complete all the planned analyses. The reviewer notes however, that the results may reflect the impact of an intervention (improved access to care) and this will be carefully discussed when the results of the cohort are shared.

VERSION 2 – REVIEW

REVIEWER	Matthew O Wiens University of British Columbia & Mbarara University of Science and Technology
REVIEW RETURNED	19-Feb-2019

GENERAL COMMENTS	Thank you for allowing me to re-review this manuscript. I have carefully read the authors responses to the previously submitted reviewer comments and I believe that all the concerns have been adequately addressed. I mentioned in my earlier review the confusing sentence (the 2nd sentence) in the "community controls" section. This sentence appears to list the inclusion criteria for community controls. It may be good to explicitly state this. The sentence as written still does not appear to be complete, and specifying who these criteria apply to would clarify this sentence.
---

VERSION 2 – AUTHOR RESPONSE

2. Reviewer 1:

A. I mentioned in my earlier review the confusing sentence (the 2nd sentence) in the "community controls" section. This sentence appears to list the inclusion criteria for community controls. It may be good to explicitly state this. The sentence as written still does not appear to be complete and specifying who these criteria apply to would clarify this sentence.

Response: Thank you for your comments. We apologize for the oversight. We have now made the correction to reflect your feedback. The amended sentence can be viewed on page 7 of the updated manuscript and now reads as follows: "To establish community norms, 125 children at each site living in the same community as hospitalised participants are recruited as community participants based on the following inclusion criteria: absence of known untreated HIV or TB; no hospital admission in the 14 days prior to contact with the study team; and no previously participation in the study."